# Psychoeducation for Relatives of Young Adults with First-Episode Psychosis: A Qualitative Exploration of Needs and Experiences

**DOI:** 10.3390/nursrep15060197

**Published:** 2025-06-03

**Authors:** S. A. Kuipers, C. A. Elzinga-Hut, B. S. Rosema, S. Sanches, D. Boertien, B. Stavenuiter, S. K. Spoelstra, G. H. M. Pijnenborg, N. Boonstra

**Affiliations:** 1Research Group Healthcare & Innovation in Psychiatry, Department of Healthcare, NHL Stenden University of Applied Sciences, Rengerslaan 8-10, 8900 CG Leeuwarden, The Netherlandskor.spoelstra@nhlstenden.com (S.K.S.); nynke.boonstra@nhlstenden.com (N.B.); 2University of Applied Science GGZ-VS, Catharijnesingel 56-1, 3511 GE Utrecht, The Netherlands; 3UMC Utrecht Brain Center, University Medical Center Utrecht, Heidelberglaan 100, 3584 CX Utrecht, The Netherlands; 4Addiction Care North Netherlands, Laan Corpus Den Hoorn 102, 9728 JR Groningen, The Netherlands; 5Patient Association Anoiksis, UMCU Kamer B.01.3.38, Heidelberglaan 100, 3584 CX Utrecht, The Netherlands; 6Department of Learning, Innovating and Co-creating (LIC), Avans University of Applied Sciences, Hogeschoollaan 1, 4818 CR Breda, The Netherlands; 7Phrenos Center of Expertise, Da Costakade 45, 3521 VS Utrecht, The Netherlands; 8Relatives Association Ypsilon, Regulusweg 5, 2516 AC Den Haag, The Netherlands; 9Department of Experimental Psychopathology and Clinical Psychology, Faculty of Behavioural and Social Sciences, University of Groningen, Grote Kruisstraat 2/1, 9712 TS Groningen, The Netherlands; g.h.m.pijnenborg@rug.nl; 10Department of Psychotic Disorders, GGZ Drenthe Mental Health Institute, Beilerstraat 197, 9401 PJ Assen, The Netherlands; 11KieN Early Intervention Service, Oosterkade 72, 8911 KJ Leeuwarden, The Netherlands

**Keywords:** nursing, first episode psychosis, psychoeducation, family-interventions, relatives

## Abstract

**Background/Objectives:** Although psychoeducation for relatives of individuals with a first episode psychosis is important for increasing understanding of psychosis, reducing relapse rates, decreasing hospitalization duration, and improving patient functionality, there is limited research on the specific experiences and needs of relatives of patients with a first episode psychosis. This study aims to explore the experiences and needs of relatives of young adults with first-episode psychosis regarding psychoeducation, with the goal of developing tailored psychoeducation (PE) that can be delivered by nurses. **Methods:** This qualitative study employed a descriptive, interpretative approach with a total sample of 23 participants, including semi-structured interviews (N = 16), two dyadic interviews (N = 4) and one triadic interview (N = 3). The dyadic interviews included two relatives and two patients, while the triadic interview involved two relatives and one patient. A topic list was utilized to guide the interviews. Thematic analysis was employed to analyse the data, supported by the use of ATLAS.ti. **Results:** During data analysis, five key themes were identified as relevant for the development of a psychoeducational program: experiences with first-episode psychosis and psychoeducation, the content of PE (what), timing (when), exchanging experiences (how) and joint PE versus separate groups (which format). **Conclusions**: This study highlights valuable insights and key components for an integrated psychoeducation program, focussing on the needs and experiences of relatives, for the development of the PE program. To optimize the benefits for both parties, future research should explore the potential of offering PE sessions that accommodate both individual and combined participant formats, allowing for a design tailored to the specific needs of the participants.

## 1. Introduction

First-episode psychosis (FEP) typically emerges during adolescence and young adulthood, a critical period for brain development [1]. Psychotic disorders at this stage can disrupt key developmental milestones, impairing both social and personal functioning [2,3], impacting not only individuals with FEP but also their relatives.

Relatives of young adults with FEP often experience anxiety, guilt, stigma, depression, and uncertainty about providing optimal support to their family members [4,5]. Many relatives lack essential knowledge about symptoms, treatment options, and the organization of the mental health care system, particularly if they have no prior experience with schizophrenia-spectrum disorders and mental health care [6]. Given these challenges, psychoeducation (PE) for relatives, in addition to PE for patients, is a crucial intervention in FEP treatment.

Over time, PE has evolved, influenced by developments in mental health care and by research on, for example, the role of relatives’ Expressed Emotions (EE) in patients with psychotic disorders [7]. PE has been empirically validated as an effective approach for enhancing the understanding of psychosis, reducing relapse rates, decreasing hospitalization duration, and improving patient functioning [8,9]. Despite these benefits, PE for relatives remains underutilized in clinical practice, often depending on individual mental health providers’ initiatives.

PE is recommended as an integral part of the treatment of patients with a psychotic disorder, including for individuals with FEP [9]. It is a low-risk psychosocial intervention that systematically provides information on diagnosis, treatment, and management. By equipping relatives with communication and problem-solving skills a supportive recovery environment can be fostered [9]. PE programs typically cover disease-specific topics such as relapse symptoms and genetic risks, along with general skills like stress management and communication training [10]. Nolan & Petrakis (2019) highlight the importance of tailoring PE to individual circumstances while providing opportunities for experience sharing [11,12]. Thus, comprehensive PE should integrate three basic components: information, skills training, and peer exchange.

The preferred timing of PE varies between patients and relatives. Studies show that relatives often seek information immediately [13], whereas patients may prefer delayed engagement due to cognitive challenges and mistrust [11]. The structure of PE programs also differs, ranging from single-family to multi-family sessions, with or without patient involvement [14,15]. Programs also vary in duration, from four sessions [16] to an average of seventeen, with maintenance sessions extending over a year [17]. Research consistently shows that structured family psychoeducation (PE), the most widely studied family intervention for psychotic disorders, leads to improved patient outcomes, including reduced relapse rates, fewer readmissions, and better medication adherence [18,19,20,21,22].

In clinical practice, as the largest group of mental health providers, nurses play a vital role in offering PE to patients and relatives [11,21]. Due to their frequent and often longstanding contact with both patients and their relatives, nurses are particularly well-suited to guide in PE. A recent study highlighted the importance of supporting mental health nurses as facilitators of PE groups through targeted education and structured program frameworks [22]. Additionally, nursing-led PE to caregivers of patients with schizophrenia has been shown to reduce family burden and enhance the quality of life of relatives [23].

Although there is evidence supporting the positive effects of PE for relatives of young adult patients with FEP, to our knowledge, no studies have specifically explored the experiences and needs of relatives of young adults with FEP regarding PE.

This study aims to explore the experiences and needs of relatives of patients with FEP concerning PE. By identifying these experiences and needs, we aim to guide the development of tailor-made PE programs for relatives of patients with FEP.

## 2. Materials and Methods

### 2.1. Study Design

A qualitative, descriptive-interpretive approach was used to gain insight into the needs and experiences of relatives of patients with FEP regarding PE.

### 2.2. Participant Selection

The study population consisted of relatives of patients with FEP (i.e., parents, siblings, and partners) (Table 1). Since the perspectives of patients are also important in the development of the PE program for relatives, and because relatives reported a preference for integrated PE with both relatives and patients, individuals with an FEP within the past five years were also consulted. Participants were recruited through the Dutch Network for Early Psychosis of which 28 teams across the Netherlands are members. Recruitment was done by its website and by email invitations sent to members of this network of mental health care professionals, who then informed relatives and patients about the opportunity to participate in the study. All participants signed up for an interview by sending an email to the interviewers.

Relatives were included if they were involved in the care and support of individuals with FEP, including family members, and 18 years or older. Relatives were excluded if they did not speak the Dutch language.

Patients were included if their FEP occurred before the age of 35 and they had received treatment for this condition. Patients were excluded if they experienced an active psychotic episode during the study. A convenience sample was used, based on participants’ availability and willingness to participate.

### 2.3. Data Collection

The interviews were conducted between May and November 2022. Initially, individual semi-structured interviews explored the needs and experiences of patients with FEP and their relatives. To deepen insights, two dyadic and one triadic interview followed, allowing for experience-sharing and open conversations [24].

Interviews began with an introduction of the researchers, followed by the open-ended question: “What is your experience with first-episode psychosis and PE?”. An interview guideline was utilized, based on the topics identified in the literature. The topics included experiences with PE, needs of relatives and patients regarding PE, required information and skills regarding PE, and timing of PE. Subsequent questions were guided by participants’ responses.

After obtaining informed consent, individual interviews were conducted by a trained nurse-researcher (C.E.), either online or in person at the care facility, depending on participant preference. Individual interviews lasted 30–60 min, while dyadic and triadic interviews lasted 60–120 min, facilitated by a nurse-researcher (C.E.) and a researcher in experiential knowledge and recovery support (D.B.).

Sampling, data collection, and analysis continued iteratively until data saturation was reached, with no new codes emerging in the final two interviews with relatives. Field notes were taken during and after interviews. Verbatim transcripts were generated [25], after which videotapes and audiotapes were deleted to ensure anonymity.

### 2.4. Data Analysis

The data from the individual interviews were analysed and coded by a member of the research team (C.E.), while the data from the dyadic and triadic interviews were analysed and coded independently by two members of the research team (C.E. & D.B.). We analysed the data using both a deductive and inductive thematic analysis approach. The three phases for a thematic analysis were used for analysis [26]. First, structures and patterns emerged through fragmentation, with the fragments being assigned open codes. Second, during the reduction phase, coherent themes were identified based on these codes. These themes were subsequently revised and refined. In the third phase, we reflected on the themes and the analytical process, in which the themes have been further refined [26,27].

During this analytic process, peer debriefing [28] facilitated the accuracy and credibility of the study by identifying any gaps and inconsistencies, judgments, biased views, and assumptions or biases in the data. Peer debriefing was carried out with four other team members (B.S.R., B.S., M.P., N.B.). All team members contributed from a different perspective; B.S.R. represented the patient perspective, B.S. the relatives’ perspective; M.P. the scientific and professional perspective from a psychological point of view and N.B. the scientific and professional perspective from a nursing point of view. The key themes derived from the analysis are presented in the result section and illustrated with quotes from participants. By aiming to accurately represent participants’ needs, authenticity was ensured. In this study, credibility was established through participant validation; to explore the credibility of the results, verbatim transcripts were returned to participants of the individual interviews [25]. To explore the credibility of the results in the dyadic/triadic interviews, a summary of the interviews was sent to participants. All data were analysed using Atlas TI version 24 (atlasti.com accessed on 9 September 2024).

### 2.5. Ethical Considerations

Participants were, prior to the interview, provided with oral and written information about the research, including details on confidentiality, voluntary participation, guaranteed anonymity, and the possibility to withdraw from the study at any time. Written informed consent was given before the start of the interviews. The transcripts of the interviews were stored in accordance with the international safety regulations for the storage of data and are anonymized and securely stored in Data Stations Social Sciences.

## 3. Results

In total, twenty-three participants took part in the study, including relatives (n = 20) and patients (n = 3). The relatives participated in sixteen individual interviews, while the remaining seven participants were involved in dyadic/triadic interviews. The 16 individual semi-structured interviews included nine parents, one partner and six siblings. The first interview was a triadic interview and consisted of two relatives and one patient and focused on initial ideas and experiences about PE. The second interview was a dyadic interview with two relatives, and the third was a dyadic interview with two patients. The patients in the study had an age of onset of FEP ranging from 18 to 34 years. All participants were of European descent. Table 1 presents the demographics of all participants, gender, age of onset FEP and relationship towards the patient. Table 2 provides an example of the analysis process using the steps of a thematic analysis.

During data analysis, five key themes were identified as relevant for the development of a PE program: experiences with FEP and PE, content of PE (*what*), timing (*when*), exchanging experiences (*how*) and joint PE versus separate groups (*which format*).

### 3.1. Experiences with FEP and PE

Some relatives reported having received information about FEP, primarily regarding what FEP is and the nature of psychosis. Half of the relatives reported receiving no information at all, which they perceived as a substantial shortcoming. The uncertainty about what was happening caused anxiety among relatives, and this fear persisted due to the lack of information. Through PE, parents can not only gain practical knowledge about FEP but also learn how to cope with the emotional impact of such situations, helping them feel heard and involved.

“*I am a mother, and for that reason, I became involved in the healthcare system—‘if you can’t beat them, join them.’ I received no explanation whatsoever. My son was 17 when he was admitted, which I found devastating. He was extremely frightened, overpowered, and placed in the forensic psychiatric center. I was not allowed to visit him because I first had to be screened. As a mother, I felt unheard and unseen.*”(P17)

“*Initially, the PE sessions concentrated on elucidating the nature of psychosis, with practical considerations regarding its management introduced at a later stage. During a group session, there was recognition and exchange of experiences, which led to several questions being discussed, e.g., “How long will this persist? Will it improve? What steps can be taken to facilitate recovery and enhance daily life?*”(P4)

In contrast to relatives, patients (n = 3) reported that their relatives had received individual PE. This involved parents and a partner. One patient mentioned that her parents had attended PE sessions over four meetings, which was a positive experience that improved their relationship.

“*The PE has had a positive effect on my parents. I now have a much better relationship with them. Before the psychosis, I struggled a lot with them. They were constantly on top of me. I suspect that through the PE, they learned to give me more space. That has been a huge relief. In that sense, the psychosis should have happened earlier. *”(P23)

All patients stated that a reader with information was provided for further review at home. While providing information to family members can be beneficial, it is important that PE is structured in a way that fosters inclusivity and empowerment rather than reinforcing feelings of exclusion. However, one patient found it uncomfortable that the PE was individually targeted at relatives.

“*My partner occasionally mentioned that he benefited from it. However, the way he expressed it felt belittling to me—as if I was the one who was sick, and he had to take care of me. I felt very excluded because I had no idea what was being shared about me or my illness. The PE for my partner was about psychosis, not about me, let alone involving me.*”(P23)

### 3.2. Content of Psychoeducation

Within the theme of content, three distinct subthemes were identified: (1) the need for knowledge, (2) the need for skill development, and (3) online information.

#### 3.2.1. Need for Knowledge

All relatives reported the importance of gaining general knowledge about psychosis. Topics they identified as important included signs & symptoms of psychosis and relapse, prognosis (including recovery and adding that recovery takes time) and treatment, particularly the effects of medication and lifestyle interventions.

“*When my son was hospitalized, I first wanted to know basic information about psychosis. What were the signs leading up to the episode? How long does it take to recover? What can we expect for the future?*”. “*After the first information about psychosis, questions arise like: how long will this take? Will it ever be okay again? What can we do to make life easier?*”(P4)

Almost all relatives also shared the confusion they experienced when first being introduced to the mental health care system. Relatives expressed a need for information about how the system works, the meaning of all abbreviations and how to access the help they need within this system. For example, relatives indicated that some kind of flow diagram could be helpful.

In addition to the importance of sufficient knowledge for relatives, they also indicated that it is essential for this information to be available online. It would be valuable to supplement the information with videos and personal stories, allowing relatives to learn in an accessible way and feel supported by the personal experiences of others.

One patient mentioned receiving basic information about psychosis during their treatment but expressed a desire to revisit this basic information within PE. Patients also wanted to share their personal experiences of what signs and symptoms matched their specific situation. For example: while some patients with psychosis were experiencing anxiety during the psychosis, others may feel in control of the world and feel unusually empowered. One patient (P23) emphasized the importance of helping relatives recognize early signs of relapse specific to their situation and to learn that every psychosis is different. She stated that it is important to provide relatives with tips on proper self-care following psychosis: physical activity, nutrition, and sleep. This will help relatives to recognize when someone is beginning to neglect themselves and understand what actions they can take.

“*I think it is important to provide relatives with guidance on proper self-care following a psychotic episode, including physical activity, nutrition, and sleep. This will help them recognize signs of self-neglect and understand how to respond appropriately.*”(P23)

#### 3.2.2. Need for Skill Development

After the basic information on psychosis, relatives perceived communication with patients as often leading to a struggle between reality and psychosis. They sought a way to engage in conversation without triggering an immediate conflict. Relatives emphasized the importance of receiving practical skills and tips for this, as well as exercises in communication skills. Additionally, relatives as well as siblings prefer learning problem-solving skills. Both consider it important to receive guidance on how to best interact with someone when a FEP has occurred. Several participants indicated that they had to figure it out on their own, which was perceived as unpleasant.

“*I wanted to know how to interact with my child who experienced psychosis, how do I maintain contact.*”(P1)

“*Practical tips on how to communicate in a way we don’t end up in a fight but to stay in control as parents. In a way, we both still have energy after we spoke to one another. And the feeling we solved something.*”(P3)

Specifically, ‘connecting’ conversation skills were mentioned by different relatives. Connecting conversation skills included how to communicate your own needs as a relative as well as having an open and nonjudgemental attitude towards the patient.

“*Eventually we asked for conversation skills that would benefit the connection with our child instead of only facing the problems. This was the first thing that really helped us out.*”(P3)

Some relatives mentioned stigma and emphasized its importance as part of PE, particularly regarding when and to whom to disclose such experiences. They emphasized that understanding how to navigate disclosure and cope with stigma should be an important component of PE.

“*Our son had a very negative experience when sharing his story in the student house where he was living at the time. When he first started studying, he did not yet have a close group of friends, which made it very lonely. When he eventually decided to share his experience, he was no longer allowed to stay in the student house.*”(P17)

All patients highlighted the importance of discussing stigma and carefully considering when and with whom information is shared. The focus should not only be on explaining the illness and its disclosure but also on preventing setbacks and maintaining social networks. The stigma related to psychosis is profound, and well-structured PE can help alleviate much of this distress. Additionally, patients state that the composition of their support network is vital:

“*Having “healthy” individuals around who can provide positive inspiration and encouragement is valuable. The people included in the network can make a significant difference.*”(P22)

#### 3.2.3. Need for Reliable Online Information

Relatives were asked how they obtained their information, if not provided through mental health care. All relatives started searching online for information on psychosis. However, most of the participants were in doubt about the reliability of the information. Therefore, they express the need for a list of reliable websites for information in PE.

“*You can Google something like “what is psychosis”, read it and think that is the truth. But the website can describe psychosis way worse than it is in real life. I would have liked a list with reliable websites where I can read the information knowing it is valid.*”(P11)

### 3.3. Timing Psychoeducation

Almost all relatives stated they strongly prefer PE as soon as possible after the onset of FEP. Because of the unfamiliarity with psychosis and the mental health care system, the need for information was high right from the very beginning.

“*I think within the first month would be good. It doesn’t have to be on the first day.*”(P6)

Half of the relatives also mentioned it was challenging for them to obtain information. The mental health care professionals did not take the initiative to provide PE for the relatives. However, relatives received proper information when they asked for it themselves.

“*We had an initial interview with the psychiatrist, but we had to ask for this ourselves. Our questions were only answered when requested by ourselves. But when we did ask our questions, the professional would take the time to answer them.*”(P2)

Patients, on the other hand, reported they had to stabilize first before they were ready to receive PE. Initially, they reported problems with concentration and could be suspicious about the information provided in the beginning. Therefore, they preferred to receive PE at a later stage.

“*The timing of PE matters a lot. I was not able to receive information while I was still in my psychosis. I needed to stabilize first before I could handle it.*”(Participant 22)

### 3.4. Exchanging Experiences

Parents indicated that they benefitted from sharing experiences with other parents in similar situations. However, siblings reported a different perspective. They discovered that talking directly with the patient helped them better understand their siblings’ experiences than just sharing information with other siblings of patients with FEP. Nevertheless, siblings acknowledged the potential advantages of multi-family groups for sharing experiences. Including a (family) expert by experience could help relatives with an opportunity to ask questions and gain insights from someone with firsthand knowledge of similar situations.

When patients and relatives were asked about their preferences for PE, all participants favoured unanimously face-to-face PE sessions. Relatives emphasized that they find it difficult to build confidence and trust within the group in online sessions, and this would negatively impact the process of exchanging experiences.

“*I don’t think it is possible to gain trust in the other group members in online sessions. I need face-to-face contact to get to know the other group members and build trust to exchange experiences.*”(P11)

Relatives stated they needed support to take care of themselves as caregivers. They reported that guidance on self-care is an important item in PE.

“*What I really missed with regards to psycho education is what you would need as an informal caregiver. Some kind of coach that guides you through the world of mental health care.*”(P13)

### 3.5. Joint PE Versus Separate Groups

Some relatives had participated in multi-family PE groups (without patients) and benefitted from the opportunity to exchange experiences with other relatives. Almost all relatives expressed that they would prefer a joint PE program with the patient.

“*We had PE as a family. But I would have preferred something more like family-coaching and how to deal and communicate with the patient together. More than just giving information.*”(P10)

Almost all relatives expressed optimism about the potential benefits of a joint PE program, believing it could help patients feel less alone and give relatives hope through shared experiences. However, the need for flexibility was emphasized by relatives, as joint coping should be tailored to each patient’s-relatives unique situation and dynamics rather than following a standard approach.

“*I have no experience with PE groups that have participants with mixed roles, but I can see how it could benefit everybody. Patients get the chance to see they are not alone, and relatives can get hope from seeing how other patients and relatives get through this period.*”(P20)

“*I don’t think you can make a standard approach on how to deal with a patient and a relative. The joint-coping must be discussed with the patient to see if it works for his/her situation.*” (P7)

Patients clearly advocated for a joint PE program with their relatives. They reported the importance of a joint PE program to share their experiences of psychosis and collaboratively learn effective communication strategies.

“*PE must be with me and about me, not without me and about me. I want to let my relatives know what works for me and what signs and symptoms apply to me or don’t. It would be helpful when my relatives could see the first signs of relapse. (…) If I can tell them myself in the PE* “*When a and b happen, do this and that*”, *it could prevent a hospitalization.*”(P23)

“*We never received PE together. It was indeed not very recovery-oriented and certainly not hopeful.*”(P22)

## 4. Discussion

This study is, to the best of our knowledge, the first to explore the PE experiences and needs of relatives of young adults with FEP.

In line with previous research on PE, both relatives and patients agreed on the key content needed in PE, such as recognizing signs & symptoms of psychosis and relapse, understanding stigma, prognosis, and treatment options, including the effect of medication and lifestyle interventions [11,18]. Additionally, recent literature on the lived experiences of family members and caregivers of individuals with psychosis highlight several important themes for relatives, such as the search for an explanation for the psychotic experiences, navigating in fragmented healthcare systems, and coping with feelings of invisibility while striving to become active partners in care [29]. Other challenges identified are difficulties in communicating with the affected person, fighting stigma and isolation, dealing with uncertainty about the future, and working to build resilience and maintain hope [29]. Our findings show the need to tailor information in a PE, offer targeted skills training and provide structured opportunities for experience sharing.

A key finding in this study is that relatives are, on the one hand, eager to exchange experiences among themselves, yet at the same time they are primarily concerned with the question: how do I maintain contact with my loved one? Patients, in turn, experienced the “PE for relatives” approach as paternalistic. Combined with the positive recommendations from both the literature and practice regarding multifamily groups, this has led to a deliberate choice for a mixed approach. This choice also aligns with the recognized need for polyphony in recovery-oriented support [30]. This is hopeful because Lincoln et al. (2007) found that especially combined PE interventions for patients and their relatives have a positive impact on preventing relapse [31]. These findings suggest that a joint-focused approach to psychoeducation could enhance its relevance and impact.

An important factor for the success of joint PE is appropriate timing. In our study, relatives expressed a preference for PE early in the process, while patients indicated that they first wanted to focus on recovery. However, during the acute phase, patients often faced difficulties in engaging in PE due to impaired attention and concentration, although they still preferred joint sessions with their relatives [13]. Leonhardt et al. (2020) stated that patients’ clinical insight and metacognition are often comprised during the acute phase, which explains patients’ inability to participate fully in PE at that time [32]. The current study shows that the content of the program and the timing of the PE should be tailored to the specific needs of both relatives and patients. It also emphasizes that the optimal timing for PE may differ between the two groups. This suggests that nursing group leaders should not deliver a fixed program but instead adopt a flexible and responsive approach—one that allows them to zoom out, assess individual needs, listen attentively, and respond sensitively to the moment. Such an approach requires both a shift in attitude and the development of specific skills by those delivering psychoeducation. One model that reflects this mindset is Peer-supported Open Dialogue (POD) [33]. POD integrates peer support and Open Dialogue, offering a framework that aligns closely with the principles of flexibility, shared decision-making, and relational care. Peer support involves individuals drawing on their own lived experience of recovery to foster understanding, connection, and hope in others [34]. Open Dialogue emphasizes open, inclusive conversations between patients, family members, and professionals, with decisions made collaboratively and support provided without delay. As such, POD supports the active engagement of patients and their social networks, underlining the importance of recognizing and strengthening the role of peer workers in therapeutic collaboration [34,35]. Although PE and POD are approached with different objectives, integrating POD into PE might create a powerful synergy. This approach ensures that PE is not only informative but also interactive, experience-based, and recovery-oriented.

Although joint psychoeducation offers clear advantages, findings from the current study suggest that postponing support for relatives until patients are ready may not be optimal. When relatives receive information in advance, patients may become distrustful or feel disadvantaged, as it can create the impression that discussions are taking place about them rather than with them. An alternative approach could involve offering relatives a preparatory session with the treating clinician during the acute phase [11]. In this context, relatives may be given preliminary access to information—on the condition that the same content will later be addressed within the group setting. Clarifying to patients the rationale for early information-sharing with relatives is essential to support relatives’ understanding and engagement, thereby strengthening their ability to contribute meaningfully to the shared recovery process. Nilsen et al. (2014) concluded that meeting others in similar situations reduced feelings of shame and increased hope for the future among relatives [13]. Listening to real-life stories was perceived as more important than lectures and workshops [13]. The findings suggest that giving the group autonomy in determining the session sequence can enhance the relevance and responsiveness of the intervention. This insight should be integrated into the development of PE.

Our findings highlight that both patients and relatives struggle to integrate the received information when PE is delivered separately. Communication skills in partial are difficult to develop without the input of both patients and relatives. A risk is that relatives may become emotionally over-involved, which underscores the importance of carefully considering the timing and content of joint PE [36].

Previous research has shown that high levels of critical comments, hostility and emotional over-involvement among family members (commonly known as high expressed emotion, EE), are associated with poorer patient outcomes, including more frequent relapses and hospitalizations [37,38,39]. High expressed emotion is also associated with an increased care burden and less effective coping strategies among relatives [40,41]. Family interventions for psychosis have been shown to reduce EE [42] and alleviate the care burden for relatives. Although providing PE exclusively to parents also leads to a reduction in EE, the findings strongly support the integration of both relatives and patients in PE, as it can ease the burden on both parties and improve treatment outcomes for the patient.

### 4.1. Limitations

Our study has several limitations. Despite efforts to include diverse participants from various socio-economic and ethnic backgrounds across the Netherlands, we were unable to recruit participants from non-Western ethnic groups. This may impact findings, as psychosis is understood differently across cultures, suggesting that PE content may need to be adapted for these groups.

While dyadic and triadic interviews offer valuable insights, there is a risk of one participant dominating the discussion, creating power imbalances [43]. Although data saturation was achieved, the small sample size—particularly in dyadic and triadic interviews—and the use of convenience sampling may have introduced selection bias, limiting the diversity of perspectives and the generalizability of the findings. Consequently, the conclusions should be interpreted with care. Notwithstanding these limitations, this study yields meaningful insights into the psychoeducational needs of relatives and provides a robust basis for future program development.

### 4.2. Future Research

To build on our findings, future studies should incorporate a quantitative phase that considers the psychoeducational needs and priorities of both relatives and patients.

Future research should include more diverse cultural groups to ensure that the PE content is appropriately tailored to different backgrounds.

Long-term studies are needed to assess the impact of PE on relapse rates, family satisfaction, family dynamics, psychological benefits for relatives, and patient recovery. Replicating this research with a larger and more representative sample is recommended to validate and expand upon these findings. Additionally, exploring the benefits of joint PE programs for patients and relatives, as well as personalized approaches, could improve engagement and outcomes.

Future research should specifically investigate the role of mental health nurses in delivering PE, examining their impact on reducing highly expressed emotion (EE) and enhancing coping strategies among patients and families.

Furthermore, future nursing research should explore how integrating POD into PE can make psychoeducation more interactive, experience-based, and recovery-oriented. Additionally, studies should explore how PE, including POD, can be effectively integrated into nursing practice and established as a fundamental component of mental health nursing education and professional development.

This could help bridge the information gap for patients, relatives, and healthcare providers. Additionally, it is important to gain insight into the barriers and facilitators for delivering PE. The potential of technological solutions, such as online platforms should also be further explored [44]. The benefits of PE extend beyond reducing relapse rates. A combined approach may also enhance family and patient satisfaction and strengthen the quality of collaboration within the family. To build on these insights, future research should further examine the effectiveness of PE in improving these aspects. Meanwhile, the findings from this study are already being translated into concrete applications, including a practice-oriented psychoeducation module for relatives and patients, a conversation guide, and workshops for professionals (https://kenniscentrumphrenos.nl, accessed on 20 March 2025).

## 5. Conclusions

This study highlights valuable insights and key components for the development of the PE program. The findings underscore the importance of tailoring psychoeducational content to include not only factual knowledge but also communication skills and access to trustworthy online resources. Moreover, the timing of psychoeducation should be adapted to the differing trajectories of patients and relatives throughout the recovery process. The exchange of experiences was identified as a valuable component for both groups, reinforcing the relational and experiential aspects of recovery. Notably, the concept of POD and a joint psychoeducational program was viewed as a promising avenue to enhance understanding and strengthen support networks. These insights provide a strong foundation for the development of a PE program, allowing for a design tailored to the specific needs of the participants.

## Figures and Tables

**Table 1 nursrep-15-00197-t001:** Participant demographics (gender of relative, age of onset FEP and relationship towards patient) (N = 23).

Participant Number	Gender of Relative	Age of Onset of FEP for Patient	Relation Towards Patient
1	Female	29	Parent
2	Female	22	Parent
3	Female	18	Parent
4	Female	24	Parent
5	Female	25	Parent
6	Female	22	Sister
7	Female	23	Partner
8	Male	18	Sister
9	Female	27	Brother
10	Male	25	Brother
11	Male	25	Brother
12	Female	25	Parent
13	Male	25	Parent
14	Male	25	Brother
15	Male	15	Partner
16	Female	21	Parent
17	Female	17	Parent
18	Female	20	Parent
19	Female	17	Patient
20	Female	20	Parent
21	Female	32	Parent
22	Male	28	Patient
23	Female	28	Patient

Note: P17-P23 were part of dyadic/triadic interviews.

**Table 2 nursrep-15-00197-t002:** An example of the analysis process using the steps of a thematic analysis.

Quotes	Codes	Themes
“We noticed that our son was doing worse, but we didn’t know what was going on. We also had no knowledge or experiences with the mental health care system”. (P2) “My son was admitted, and I had no idea what was happening. As his mother, I wasn’t involved in anything. It wasn’t until six days after his admission that I received the first bit of information”. (P18)	The family notices No experiences in mental health care No involvement Information after several days	Experiences with FEP and PE
“I wanted to know something about the cause of psychosis and what the effect is of psychosis on the brain, what happens during psychosis? And how do you deal with someone with psychosis?” (P11) “After the information about psychosis, questions arise like: how long will this take? Will it ever be okay again? What can we do to make life easier?” (P4)	Basic knowledge on psychosis is important Information about recovery Stigma	Content—knowledge
“Tips. Practical tips on how to communicate in a way we don’t end up in a fight but to stay in control as parents. In a way we both still have energy after we spoke to one another. And the feeling we solved something” (P3) “How can I communicate with my brother?” (P11)	Dealing with psychosis—communication Dealing with psychosis	Content—skills
“You can Google something like “what is psychosis”, read it and think that is the truth. But the website can describe psychosis way worse than it is in real life. I would have liked a list with reliable websites where I can read the information knowing it is valid.” (P11) “I think it could be beneficial for people to receive extra information prior to the psychoeducation. Before you start searching on your own and find the worst things on the internet” (P 10)	Reliability online information Online information in psychoeducation Online information and assessing quality	Content—online information
“The timing of PE matters a lot. I was not able to receive information while I was still in my psychosis. I needed to stabilize first before I could handle it” (P22) “Well, you should not wait too long [with psychoeducation]. It doesn’t need to be the next day, but you also shouldn’t wait for months” (P2)	Psychoeducation after stabilization Timing of psychoeducation for relatives quick	Timing
“I might have liked peer contact. The feeling that you’re not alone and when other people would’ve told me how long it takes and that everything will be all right… that would’ve comforted me” (P8) “Our son has a friend with the same problems, and we visited those parents a couple of times. That was nice, we could discuss things together” (P12)	Exchanging experiences is important Involve an expert by experience	Exchanging experiences
“Sometimes it could be nice when the children (patients) aren’t in the room, and you get to say things you wouldn’t dare to when they are present” (P16) “I would say that a combined psychoeducation with the patient would be nice, since you both get the same information. (…) I can imagine there are situations you would rather not do with the patient, but in terms of openness and transparency I would prefer doing it together” (P7)	Preference for joint psychoeducation Joint psychoeducation with moments apart	Joint vs. separate groups

## Data Availability

The data presented in this study are available on request from the corresponding author due to privacy and ethical restrictions.

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
