# Peer review of "Psychoeducation for Relatives of Young Adults with First-Episode Psychosis: A Qualitative Exploration of Needs and Experiences"

_nursrep, 2025, doi:10.3390/nursrep15060197_

Round 1
Reviewer 1 Report
Comments and Suggestions for Authors
This study explains the importance of exploring the experiences and psychoeducational needs of the relatives of young adults with first-episode psychosis, and then goes on to do so. In this regard, the study is coherent with an Introduction, Methods, Results and Discussion. However, I question the real contribution to the literature due to the qualitative nature of the study. It would have been more interesting to have seen what was shown as the first part in exploring the research space, and then to more quantitatively focus on particular psychoeducational features.
The conclusion is too vague and even given the qualitative nature of the study should be a bit more specific and actionable.
Nonetheless, the topic is important and the nursing psychoeducational focus is welcome.
I would add that, in any scientific field--whether medicine, physics or engineering for example--scientific inquiry typically involves hypotheses, data, and at least some degree of quantitative analysis, even if exploratory.
My enhanced review following your suggestions (Word file also attached with the same):
This study explains the importance of exploring the experiences and psychoeducational needs of the relatives of young adults with first-episode psychosis and then goes on to do so.
Main question addressed by the research: This work explores the experiences and psychoeducational needs of relatives of young adults with first-episode psychosis. In doing so, the study wants to be able to deliver better care to the family, i.e., develop psychoeducational programs (PE) that allow joint family-patient engagement.
Originality and relevance to the field: The approach to first-episode psychosis has greatly matured over the course of my career in seeing these patients (North America). This study contributes further to addressing the subjective experiences and preferences of relatives of young adults with first-episode psychosis, so it does have relevance. The nursing approach and the exploration of joint versus separate psychoeducation approaches is also relevant, and combined with the other aspects of the paper demonstrates some originality.
Contribution to the subject area compared with prior work and suggestions of actionable improvements: My background is more familiar with the technical aspects of treating first-episode psychosis than the broad nursing literature. I rely on the paper for the latter. However, I do have experience with real-world implementation of nursing care of patients and their families with this condition.
This is where I question whether the contribution to the literature is as strong as it could be, given the study’s very qualitative nature. While exploration of the topic is important to the field, I believe it would have been more advantageous and impactful to follow the qualitative exploration with a quantitative phase considering psychoeducational features and caregiver and patient priorities.
The conclusion is too vague and, even given the qualitative nature of the study, should be somewhat more specific and actionable. I would encourage the authors to suggest how to operationalize some of the findings (comments?) of the paper. Also, I would like to see a clearer distinction between what the current data show and what is proposed for future work. Also, I would like to see a more analytical additional table forcing the authors to more analytically organize and compare their ideas.
Nonetheless, the topic is important, and the nursing focus is welcome. I am hopeful that my comments above will contribute to a stronger paper that makes a better contribution to the field.

Author Response
Dear Reviewer,
Please see the attachment.
Kind regards,
Sonja Kuipers

Reviewer 2 Report
Comments and Suggestions for Authors
Dear Author(s),
Thank you for the opportunity to review this manuscript. It presents an interesting and relevant contribution; however, to enhance its academic quality and clarity, I respectfully submit the following observations and suggestions:
- Line 40: When abbreviations or acronyms are introduced, please ensure their full meanings are provided to aid reader comprehension.
- Abstract: The methodological description in the abstract is limited. It would be beneficial to include information regarding the sample size and the instrument employed (e.g., ATHAS-TI).
- Although the study is based on a qualitative approach, the presentation of data could be improved by incorporating visual aids. These may include bar charts, Sankey diagrams, donut charts, treemaps, networks, word clouds, or tables. Such additions would enrich the theoretical discussion and help prevent a monotonous narrative.
- Line 351: Please review this line for spelling or typographical errors.
- The manuscript demonstrates a strong and comprehensive review of the relevant literature, which effectively supports the findings presented.
- Lines 371–375: The content of these lines would be more appropriately placed in the Conclusion section. Please consider relocating them accordingly.
- Lines 441–445: It would be appropriate to cite a relevant literary source or scholarly reference to support the findings described in this section.
- The sample size is relatively small, which may limit the generalizability and robustness of the conclusions. This should be acknowledged or addressed in the discussion.
- Line 486: Please revise this line to eliminate redundancy and improve clarity.
I hope these suggestions contribute positively to the improvement of the manuscript.
Kind regards,
Author Response

(The authors gave the same response as above.)

Round 2
Reviewer 2 Report
Comments and Suggestions for Authors
Dear Authors;
After a thorough review of the suggestions provided, it is evident that they have been carefully considered and incorporated, resulting in a notable improvement in the paper's quality. Excellent work has been done in this regard.
Best regards.